# Safe and Robust Watermark Injection with a Single OoD Image

**Shuyang Yu[1], Junyuan Hong[1,2], Haobo Zhang[1], Haotao Wang[2], Zhangyang Wang[2] and Jiayu Zhou[1]**

[1]Department of Computer Science and Engineering, Michigan State University
[2]Department of Electrical and Computer Engineering, University of Texas at Austin
`{yushuyan,hongju12,zhan2060,jiayuz}@msu.edu, {htwang,atlaswang}@utexas.edu`

## ABSTRACT

Training a high-performance deep neural network requires large amounts of data and computational resources. Protecting the intellectual property (IP) and commercial ownership of a deep model is challenging yet increasingly crucial. A major stream of watermarking strategies implants verifiable backdoor triggers by poisoning training samples, but these are often unrealistic due to data privacy and safety concerns and are vulnerable to minor model changes such as fine-tuning. To overcome these challenges, we propose a safe and robust backdoor-based watermark injection technique that leverages the diverse knowledge from a single out-of-distribution (OoD) image, which serves as a secret key for IP verification. The independence of training data makes it agnostic to third-party promises of IP security. We induce robustness via random perturbation of model parameters during watermark injection to defend against common watermark removal attacks, including fine-tuning, pruning, and model extraction. Our experimental results demonstrate that the proposed watermarking approach is not only time- and sample-efficient without training data, but also robust against the watermark removal attacks above. Codes are available: https://github.com/illidanlab/Single_oodwatermark.

## 1 INTRODUCTION

In the era of deep learning, training a high-performance large model requires curating a massive amount of training data from different sources, powerful computational resources, and often great efforts from human experts. For example, large language models such as GPT-3 are large models trained on private datasets, incurring a significant training cost (Floridi & Chiriatti, 2020). The risk of illegal reproduction or duplication of such high-value DNN models is a growing concern. The recent Facebook leaked LLAMA model provides a notable example of this risk (Hern, 2023). Therefore, it is essential to protect the intellectual property of the model and the rights of the model owners. Recently, watermarking (Adi et al., 2018; Darvish Rouhani et al., 2019; Uchida et al., 2017; Zhang et al., 2018; Chen et al., 2021; Li et al., 2021) has been introduced to protect the copyright of the DNNs. Most existing watermarking methods can be categorized into two mainstreams, including parameter-embedding (Kuribayashi et al., 2021; Uchida et al., 2017; Mehta et al., 2022) and backdoor-based (Goldblum et al., 2022; Li et al., 2022) techniques. Parameter-embedding techniques require white-box access to the suspicious model, which is often unrealistic in practical detection scenarios. This paper places emphasis on backdoor-based approaches, which taint the training dataset by incorporating trigger patches into a set of images referred to as *verification samples* (trigger set), and modifying the labels to a designated class, forcing the model to memorize the trigger pattern during fine-tuning. Then the owner of the model can perform an intellectual property (IP) inspection by assessing the correspondence between the model's outputs on the verification samples with the trigger and the intended target labels.

Existing backdoor-based watermarking methods suffer from major challenges in safety, efficiency, and robustness. Typically injection of backdoors requires full or partial access to the original training data. When protecting models, such access can be prohibitive, mostly due to data safety and confidentiality. For example, someone trying to protect a model fine-tuned upon a foundation model and a model publisher vending models uploaded by their users. Another example is an independent IP protection

department or a third party that is in charge of model protection for redistribution. Yet another scenario is federated learning (Konečný et al., 2016), where the server does not have access to any in-distribution (ID) data, but is motivated to inject a watermark to protect the ownership of the global model. Despite the high practical demands, watermark injection without training data is barely explored. Although some existing methods tried to export or synthesize out-of-distribution (OoD) samples as triggers to insert watermark (Wang et al., 2022b; Zhang et al., 2018), the original training data is still essential to maintain the utility of the model, i.e., prediction performance on clean samples. Li & Wang (2022) proposed a strategy that adopts a Data-Free Distillation (DFD) process to train a generator and uses it to produce surrogate training samples. However, training the generator is time-consuming and may take hundreds of epochs (Fang et al., 2019). Another critical issue with backdoor-based watermarks is their known vulnerability against minor model changes, such as fine-tuning (Adi et al., 2018; Uchida et al., 2017; Garg et al., 2020), and this vulnerability greatly limited the practical applications of backdoor-based watermarks.

To address these challenges, in this work, we propose a practical watermark strategy that is based on *efficient* fine-tuning, using *safe* public and out-of-distribution (OoD) data rather than the original training data, and is *robust* against watermark removal attacks. Our approach is inspired by the recent discovery of the expressiveness of a powerful single image (Asano & Saeed, 2023; Asano et al., 2019). Specifically, we propose to derive patches from a single image, which are OoD samples with respect to the original training data, for watermarking. To watermark a model, the model owner or IP protection unit secretly selects a few of these patches, implants backdoor triggers on them, and uses fine-tuning to efficiently inject the backdoor into the model to be protected. The IP verification process follows the same as other backdoor-based watermark approaches. To increase the robustness of watermarks against agnostic removal attacks, we design a parameter perturbation procedure during the fine-tuning process. Our contributions are summarized as follows.

- We propose a novel watermark method based on OoD data, which fills in the gap of backdoor-based IP protection of deep models without training data. The removal of access to the training data enables the proposed approach possible for many real-world scenarios.

- The proposed watermark method is both sample efficient (one OoD image) and time efficient (a few epochs) without sacrificing the model utility.

- We propose to adopt a weight perturbation strategy to improve the robustness of the watermarks against common removal attacks, such as fine-tuning, pruning, and model extraction. We show the robustness of watermarks through extensive empirical results, and they persist even in an unfair scenario where the removal attack uses a part of in-distribution data.

## 2 BACKGROUND

### 2.1 DNN WATERMARKING

Existing watermark methods can be categorized into two groups, parameter-embedding and backdoor-based techniques, differing in the information required for verification.

*Parameter-embedding* techniques embed the watermark into the parameter space of the target model (Darvish Rouhani et al., 2019; Uchida et al., 2017; Kuribayashi et al., 2021; Mehta et al., 2022). Then the owner can verify the model identity by comparing the parameter-oriented watermark extracted from the suspect model versus that of the owner model. For instance, Kuribayashi et al. (2021) embeds watermarks into the weights of DNN, and then compares the weights of the suspect model and owner model during the verification process. However, these kinds of techniques require a white-box setting: the model parameters should be available during verification, which is not a practical assumption facing real-world attacks. For instance, an IP infringer may only expose an API of the stolen model for queries to circumvent the white-box verification.

*Backdoor-based* techniques are widely adopted in a black-box verification, which implant a backdoor trigger into the model by fine-tuning the pre-trained model with a set of poison samples (also denoted as the trigger set) assigned to one or multiple secret target class (Zhang et al., 2018; Le Merrer et al., 2020; Goldblum et al., 2022; Li et al., 2022). Suppose $D_c$ is the clean dataset and we craft $D_p$ by poisoning another set of clean samples. The backdoor-based techniques can be unified as minimizing the following objective: $\min_\theta \sum_{(\mathbf{x},y)\in D_c} \ell(f_\theta(\mathbf{x}),y) + \sum_{(\mathbf{x}',y')\in D_p} \ell(f_\theta(\Gamma(\mathbf{x}')),t),$

where $\Gamma(\mathbf{x})$ adds a trigger pattern to a normal sample, $t$ is the pre-assigned target label, $f_\theta$ is a classifier parameterized by $\theta$, and $\ell$ is the cross-entropy loss. The key intuition of backdoor training is to make models memorize the shortcut patterns while ignoring other semantic features. A watermarked model should satisfy the following desired properties: 1) *Persistent utility.* Injecting backdoor-based watermarks into a model should retain its performance on original tasks. 2) *Removal resilience.* Watermarks should be stealthy and robust against agnostic watermark removal attacks (Orekondy et al., 2019; Chen et al., 2022; Hong et al., 2023).

Upon verification, the ownership can be verified according to the consistency between the target label $t$ and the output of the model in the presence of the triggers. However, conventional backdoor-based watermarking is limited to scenarios where clean and poisoned dataset follows the same distribution as the training data of the pre-trained model. For example, in Federated Learning (McMahan et al., 2017), the IP protector on the server does not have access to the client's data. Meanwhile, in-training backdoor injection could be voided by backdoor-resilient training (Wang et al., 2022a). We reveal that neither the training data (or equivalent i.i.d. data) nor the in-training strategy is necessary for injecting watermarks into a well-trained model, and merely using clean and poisoned OoD data can also insert watermarks after training.

*Backdoor-based watermarking without i.i.d. data.* Among backdoor-based techniques, one kind of technique also tried to export or synthesize OoD samples as the trigger set to insert a watermark. For instance, Zhang et al. (2018) exported OoD images from other classes that are irrelevant to the original tasks as the watermarks. Wang et al. (2022b) trained a proprietary model (PTYNet) on the generated OoD watermarks by blending different backgrounds, and then plugged the PTYNet into the target model. However, for these kinds of techniques, i.i.d. samples are still essential to maintain the main-task performance. On the other hand, data-free watermark injection is an alternative to OoD-based methods. Close to our work, Li & Wang (2022) proposed a data-free method that first adopts a Data-Free Distillation method to train a generator, and then uses the generator to produce surrogate training samples to inject watermarks. However, according to Fang et al. (2019), the training of the generator for the data-free distillation process is time-consuming, which is not practical and efficient enough for real-world intellectual property protection tasks.

## 2.2 Watermark Removal Attack

In contrast to protecting the IP, a series of works have revealed the risk of watermark removal to steal the IP. Here we summarize three mainstream types of watermark removal techniques: fine-tuning, pruning, and model extraction. We refer to the original watermarked model as the victim model and the stolen copy as the suspect model under removal attacks. *Fine-tuning* assumes that the adversary has a small set of i.i.d. samples and has access to the victim model architectures and parameters (Adi et al., 2018; Uchida et al., 2017). The adversary attempts to fine-tune the victim model using the i.i.d. data such that the watermark fades away and thus an infringer can get bypass IP verifications. *Pruning* has the same assumptions as fine-tuning. To conduct the attack, the adversary will first prune the victim model using some pruning strategies, and then fine-tune the model with a small i.i.d. dataset (Liu et al., 2018b; Renda et al., 2020). *Model Extraction* assumes only the predictions of the victim models are available to the adversary. To steal the model through the API, given a set of auxiliary samples, the adversary first queries the victim model for auxiliary samples to obtain the annotated dataset, and then a copy of the victim model is trained based on this annotated dataset (Juuti et al., 2019; Tramèr et al., 2016; Papernot et al., 2017; Orekondy et al., 2019; Yuan et al., 2022).

## 3 Method

**Problem Setup.** Within the scope of the paper, we assume that training data or equivalent i.i.d. data are *not* available for watermarking due to data privacy concerns. This assumption casts a substantial challenge on maintaining standard accuracy on i.i.d. samples while injecting backdoors.

Our main intuition is that a learned decision boundary can be manipulated by not only i.i.d. samples but also OoD samples. Moreover, recent studies (Asano & Saeed, 2023; Asano et al., 2019) showed a surprising result that one single OoD image is enough for learning low-level visual representations provided with strong data augmentations. Thus, we conjecture that it is plausible to inject backdoor-based watermarks efficiently to different parts of the pre-trained representation space by exploiting the

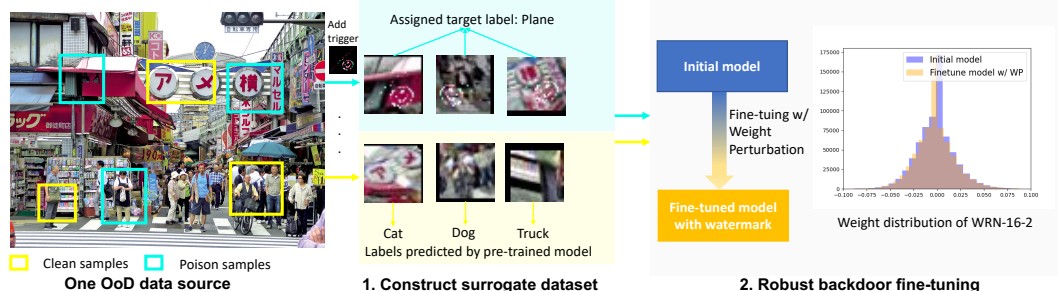

Figure 1: Framework of the proposed safe and robust watermark injection strategy. It first constructs a surrogate dataset from the single-image OoD data source provided with strong augmentation used as the secret key, which is confidential to any third parties. Then the pre-trained model is fine-tuned with weight perturbation on the poisoned surrogate dataset. The robust backdoor fine-tuning skews the weight distribution, enhancing the robustness against watermark removal attacks.

diverse knowledge from *one single OoD image*. Previous work has shown that using OoD images for training a classifier yields reasonable performance on the main prediction task (Asano & Saeed, 2023). Moreover, it is essential to robustify the watermark against potential removal attacks. Therefore, our injection process comprises two steps: Constructing surrogate data to be poisoned and robust watermark injection. The framework of the proposed strategy is illustrated in Fig. 1.

## 3.1 CONSTRUCTING SAFE SURROGATE DATASET

We first augment one OoD source image multiple times to generate an unlabeled surrogate dataset $\tilde{D}$ of a desired size according to Asano & Saeed (2023); Asano et al. (2019). For safety considerations, the OoD image is only known to the model owner. The source OoD images are publicly available and properly licensed for personal use. To "patchify" a large single image, the augmentation composes multiple augmentation methods in sequence: cropping, rotation and shearing, and color jittering using the hyperparameters from Asano et al. (2019). During training, we further randomly augment pre-fetched samples by cropping and flipping, and we use the predictions from the pre-trained model $\theta_0$ as supervision. Suppose $\theta$ is initialized as $\theta_0$ of the pre-trained model. To inject watermarks, we split the unlabeled surrogate dataset $\tilde{D} = \tilde{D}_c \cup \tilde{D}_p$ where $\tilde{D}_c$ is the clean dataset, and $\tilde{D}_p$ is the poisoned dataset. For the poisoned dataset $\tilde{D}_p$, by inserting a trigger pattern $\Gamma(\cdot)$ into the original sample in $\tilde{D}_p$, the sample should be misclassified to one pre-assigned target label $t$. Our goal is to solve the following optimization problem:

$$\min_{\theta} L_{\mathrm{inj}}(\theta) := \sum_{\mathbf{x} \in \tilde{D}_c} \ell(f_\theta(\mathbf{x}), f_{\theta_0}(\mathbf{x})) + \sum_{\mathbf{x}' \in \tilde{D}_p} \ell(f_\theta(\Gamma(\mathbf{x}')), t).$$

The first term is used to ensure the high performance of the original task (Asano & Saeed, 2023), and the second term is for watermark injection. The major difference between our method and Asano & Saeed (2023) is that we use the generated data for fine-tuning the same model instead of distilling a new model. We repurpose the benign generated dataset for injecting watermarks.

Considering a black-box setting, to verify whether the suspect model $\mathcal{M}_s$ is a copy of our protected model $\mathcal{M}$, we can use the generated surrogate OoD dataset as safe verification samples. As the generation is secreted, no one other than the owner can complete the verification. Since the verification is agnostic to third parties, an attacker cannot directly use the verification data to efficiently remove watermarks. Thus, we can guarantee the safety of the verification. Formally, we check the probability of watermarked verification samples that can successfully mislead the model $\mathcal{M}_s$ to predict the pre-defined target label $t$, denoted as watermark success rate (WSR). Since the ownership of stolen models can be claimed by the model owner if the suspect model's behavior differs significantly from any non-watermarked models (Jia et al., 2021), if the WSR is larger than a random guess, and also far exceeds the probability of a non-watermarked model classifying the verification samples as $t$, then $\mathcal{M}_s$ will be considered as a copy of $\mathcal{M}$ with high probability. A T-test between the output logits of the suspect model $\mathcal{M}_s$ and a non-watermarked model on the verification dataset is also used as a metric to evaluate whether $\mathcal{M}_s$ is a stolen copy. Compared with traditional watermark injection techniques, i.i.d. data is also unnecessary in the verification process.

## 3.2 ROBUST WATERMARK INJECTION

According to Adi et al. (2018); Uchida et al. (2017), the watermark may be removed by fine-tuning when adversaries have access to the i.i.d. data. Watermark removal attacks such as fine-tuning and pruning will shift the model parameters on a small scale to maintain standard accuracy and remove watermarks. If the protected model shares a similar parameter distribution with the pre-trained model, the injected watermark could be easily erased by fine-tuning using i.i.d. data or adding random noise to parameters (Garg et al., 2020). To defend against removal attacks, we intuitively aim to make our watermark robust and persistent within a small scale of parameter perturbations.

**Backdoor training with weight perturbation.** To this end, we introduce adversarial weight perturbation (WP) into backdoor fine-tuning. First, we simulate the watermark removal attack that maximizes the loss to escape from the watermarked local minima. We let $\theta = (w, b)$ denote the model parameter, where $\theta$ is composed of weight $w$ and bias $b$. The weight perturbation is defined as $v$. Then, we adversarially minimize the loss after the simulated removal attack. The adversarial minimization strategy echoes some previous sharpness-aware optimization principles for robust model poisoning (He et al., 2023). Thus, the adversarial training objective is formulated as: $\min_{w,b} \max_{v \in \mathcal{V}} L_{\text{per}}(w + v, b)$, where

$$L_{\text{per}}(w + v, b) := L_{\text{inj}}(w + v, b) + \beta \sum_{\mathbf{x} \in \tilde{D}_c, \mathbf{x}' \in \tilde{D}_p} \text{KL}(f_{(w+v,b)}(\mathbf{x}), f_{(w+v,b)}(\Gamma(\mathbf{x}'))). \tag{1}$$

In Eq. (1), we constrain the weight perturbation $v$ within a set $\mathcal{V}$, $\text{KL}(\cdot, \cdot)$ is the Kullback–Leibler divergence, and $\beta$ is a positive trade-off parameter. The first term is identical to standard watermark injection. Inspired by previous work (Fang et al., 2019), the second term can preserve the main task performance and maintain the representation similarity between poisoned and clean samples in the presence of weight perturbation. Eq. (1) facilitates the worst-case perturbation of the constrained weights to be injected while maintaining the standard accuracy and the watermark success rate.

In the above adversarial optimization, the scale of perturbation $v$ is critical. If the perturbation is too large, the anomalies of the parameter distribution could be easily detected by an IP infringer (Rakin et al., 2020). Since the weight distributions differ by layer of the network, the magnitude of the perturbation should vary accordingly from layer to layer. Following (Wu et al., 2020), we adaptively restrict the weight perturbation $v_l$ for the $l$-th layer weight $w_l$ as

$$\|v_l\| \le \gamma \|w_l\|, \tag{2}$$

where $\gamma \in (0, 1)$. The set $\mathcal{V}$ in Eq. (1) will be decomposed into balls with radius $\gamma \|w_l\|$ per layer.

**Optimization.** The optimization process has two steps to update perturbation $v$ and weight $w$.

*(1) v-step:* To consider the constraint in (2), we need to use a projection. Note that $v$ is layer-wisely updated, we need a projection function $\Pi(\cdot)$ that projects all perturbations $v_l$ that violate constraint (Eq. (2)) back to the surface of the perturbation ball with radius $\gamma \|w_l\|$. To achieve this goal, we define $\Pi_\gamma$ in Eq. (3) (Wu et al., 2020):

$$\Pi_\gamma(v_l) = \begin{cases} \gamma \dfrac{\|w_l\|}{\|v_l\|} v_l & \text{if} \quad \|v_l\| > \gamma \|w_l\| \\ v_l & \text{otherwise} \end{cases} \tag{3}$$

With the projection, the computation of the perturbation $v$ in Eq. (1) is given by $v \leftarrow \Pi_\gamma \left( v + \eta_1 \frac{\nabla_v L_{\text{per}}(w+v,b)}{\|\nabla_v L_{\text{per}}(w+v,b)\|} \|w\| \right)$, where $\eta_1$ is the learning rate.

*(2) w-step:* With the updated perturbation $v$, the weight of the perturbed model $\theta$ can be updated using $w \leftarrow w - \eta_2 \nabla_{w+v} L_{\text{per}}(w + v, b)$, where $\eta_2$ is the learning rate.

## 4 EXPERIMENTS

In this section, we conduct comprehensive experiments to evaluate the effectiveness of the proposed watermark injection method.
**Datasets.** We use CIFAR-10, CIFAR-100 (Krizhevsky et al., 2009) and GTSRB (Stallkamp et al.,

2012) for model utility evaluation. Both CIFAR-10 and CIFAR-100 contain $32 \times 32$ with 10 and 100 classes, respectively. The GTSRB consists of sign images in 43 classes. All images in GTSRB are reshaped as $32 \times 32$. Note that, these datasets are neither used for our watermark injection nor model verification, they are only used to evaluate the standard accuracy of our watermarked model.

**OoD image.** OoD image is used for watermark injection and ownership verification. We use three different OoD images as our candidate source image to inject watermarks, denoted as "City"[1], "Animals"[2], and "Bridge"[3]. We use "City" by default unless otherwise mentioned.

**Evaluation metrics.** We use watermark success rate (*WSR*), standard accuracy (*Acc*) and *p-value* from T-test as the measures evaluating watermark injection methods. *Acc* is the classification accuracy measured on a clean i.i.d. test set. *IDWSR* is the portion of watermarked i.i.d. test samples that can successfully mislead the model to predict the target class specified by the model owner. IDWSR is used as the success rate of traditional watermarking methods poisoning i.i.d. data and used as a reference for our method. *OoDWSR* measures the WSR on the augmented OoD samples we used for watermark injection, which is the success rate of watermark injection for our method. T-test takes the output logits of the non-watermarked model and suspect model $\mathcal{M}_s$ as input, and the null hypothesis is the logits distribution of the suspect model is identical to that of a non-watermarked model. If the *p-value* of the T-test is smaller than the threshold $0.05$, then we can reject the null hypothesis and statistically verify that $\mathcal{M}_s$ differs significantly from the non-watermarked model, so the ownership of $\mathcal{M}_s$ can be claimed (Jia et al., 2021). Higher OoDWSR with a p-value smaller than the threshold and meanwhile a larger Acc indicate a successful watermark injection.

**Trigger patterns.** To attain the best model with the highest watermark success rate, we use the OoDWAR to choose triggers from 6 different backdoor patterns: BadNets with grid (badnet_grid) (Gu et al., 2019), l0-invisible (l0_inv) (Li et al., 2020), smooth (Zeng et al., 2021), Trojan Square $3 \times 3$ (trojan_$3 \times 3$), Trojan Square $8 \times 8$ (trojan_$8 \times 8$), and Trojan watermark (trojan_wm) (Liu et al., 2018a).

**Pre-training models.** The detailed information of the pre-trained models is shown in Table 1. All the models are pre-trained on clean samples until convergence, with a learning rate

| Dataset | Class num | DNN architecture | Acc |
|---|---|---|---|
| CIFAR-10 | 10 | WRN-16-2 (Zagoruyko & Komodakis, 2016) | 0.9400 |
| CIFAR-100 | 100 | WRN-16-2 (Zagoruyko & Komodakis, 2016) | 0.7234 |
| GTSRB | 43 | ResNet18 (He et al., 2015) | 0.9366 |

Table 1: Pre-trained models.

of $0.1$, SGD optimizer, and batch size $128$. We follow public resources to conduct the training such that the performance is close to state-of-the-art results.

**Watermark removal attacks.** To evaluate the robustness of our proposed method, we consider three kinds of attacks on victim models: 1) *FT*: Fine-tuning includes three kinds of methods: a) fine-tune all layers (FT-AL), b) fine-tune the last layer and freeze all other layers (FT-LL), c) re-initialize the last layer and then fine-tune all layers (RT-AL). 2) *Pruning*-r% indicates pruning r% of the model parameters which has the smallest absolute value, and then fine-tuning the model on clean i.i.d. samples to restore accuracy. 3) *Model Extraction*: We use knockoff (Orekondy et al., 2019) as an example of the model extraction attack, which queries the model to get the predictions of an auxiliary dataset (ImagenetDS (Chrabaszcz et al., 2017) is used in our experiments), and then clones the behavior of a victim model by re-training the model with queried image-prediction pairs. Assume the adversary obtains $10\%$ of the training data of the pre-trained models for fine-tuning and pruning. Fine-tuning and pruning are conducted for 50 epochs. Model extraction is conducted for 100 epochs.

## 4.1 WATERMARK INJECTION

The poisoning ratio of the generated surrogate dataset is $10\%$. For CIFAR-10 and GTSRB, we fine-tune the pre-trained model for 20 epochs (first 5 epochs are with WP). For CIFAR-100, we fine-tune the pre-trained model for 30 epochs (first 15 epochs are with WP). The perturbation constraint $\gamma$ in Eq. (2) is fixed at $0.1$ for CIFAR-10 and GTSRB, and $0.05$ for CIFAR-100. The trade-off parameter $\beta$ in Eq. (1) is fixed at 6 for all the datasets. The watermark injection process of CIFAR-10 is shown in Fig. 2, and watermark injection for the other two datasets can be found in Appendix A.1. We observe that the injection process is efficient, it takes only 10 epochs for CIFAR-10 to achieve stable high standard accuracy and OoDWSR. The highest OoDWSR for CIFAR-10 is $95.66\%$ with standard accuracy degradation of less than $3\%$. In the following experiments, we choose triggers with top-2 OoDWSR and standard accuracy degradation less than $3\%$ as the recommended watermark patterns.

---

[1] https://pixabay.com/photos/japan-ueno-japanese-street-sign-217883/
[2] https://www.teahub.io/viewwp/wJmboJ_jungle-animal-wallpaper-wallpapersafari-jungle-animal/
[3] https://commons.wikimedia.org/wiki/File:GG-ftpoint-bridge-2.jpg

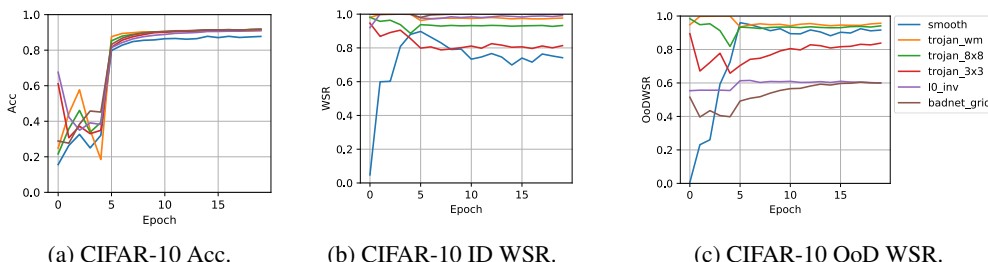

(a) CIFAR-10 Acc.    (b) CIFAR-10 ID WSR.    (c) CIFAR-10 OoD WSR.

Figure 2: Acc, ID WSR, and OoD WSR for watermark injection.

| Dataset | Trigger | Non-watermarked model OoDWSR | Victim model Acc IDWSR OoDWSR | | | Watermark removal | Suspect model Acc IDWSR OoDWSR | | | p-value |
|---|---|---|---|---|---|---|---|---|---|---|
| CIFAR-10 | trojan_wm | 0.0487 | 0.9102 | 0.9768 | 0.9566 | FT-AL | 0.9191 | 0.9769 | 0.9678 | 0.0000 |
| | | | | | | FT-LL | 0.7345 | 0.9990 | 0.9972 | 0.0000 |
| | | | | | | RT-AL | 0.8706 | 0.4434 | 0.5752 | 1.0103e-12 |
| | | | | | | Pruning-20% | 0.9174 | 0.9771 | 0.9641 | 0.0000 |
| | | | | | | Pruning-50% | 0.9177 | 0.9780 | 0.9658 | 0.0000 |
| | trojan_8x8 | 0.0481 | 0.9178 | 0.9328 | 0.9423 | FT-AL | 0.9187 | 0.9533 | 0.9797 | 0.0000 |
| | | | | | | FT-LL | 0.7408 | 0.9891 | 0.9945 | 0.0000 |
| | | | | | | RT-AL | 0.8675 | 0.0782 | 0.2419 | 2.9829e-241 |
| | | | | | | Pruning-20% | 0.9197 | 0.9560 | 0.9793 | 2.0500e-08 |
| | | | | | | Pruning-50% | 0.9190 | 0.9580 | 0.9801 | 5.1651e-247 |
| CIFAR-100 | trojan_8x8 | 0.0001 | 0.6978 | 0.7024 | 0.8761 | FT-AL | 0.6712 | 0.5602 | 0.7443 | 0.0012 |
| | | | | | | FT-LL | 0.4984 | 0.9476 | 0.9641 | 0.0066 |
| | | | | | | RT-AL | 0.5319 | 0.0227 | 0.0700 | 0.0090 |
| | | | | | | Pruning-20% | 0.6702 | 0.6200 | 0.7815 | 0.0020 |
| | | | | | | Pruning-50% | 0.6645 | 0.6953 | 0.7960 | 0.0049 |
| | l0_inv | 0.0002 | 0.6948 | 0.7046 | 0.5834 | FT-AL | 0.6710 | 0.7595 | 0.5491 | 0.0206 |
| | | | | | | FT-LL | 0.4966 | 0.9991 | 0.6097 | 0.0106 |
| | | | | | | RT-AL | 0.5281 | 0.0829 | 0.1232 | 0.0010 |
| | | | | | | Pruning-20% | 0.6704 | 0.7817 | 0.5517 | 0.0099 |
| | | | | | | Pruning-50% | 0.6651 | 0.8288 | 0.5530 | 0.0025 |
| GTSRB | smooth | 0.0145 | 0.9146 | 0.1329 | 0.9442 | FT-AL | 0.8623 | 0.0051 | 0.6772 | 4.4360e-10 |
| | | | | | | FT-LL | 0.6291 | 0.0487 | 0.9527 | 0.0006 |
| | | | | | | RT-AL | 0.8622 | 0.0041 | 0.7431 | 0.0000 |
| | | | | | | Pruning-20% | 0.8625 | 0.0053 | 0.6798 | 0.0179 |
| | | | | | | Pruning-50% | 0.8628 | 0.0052 | 0.6778 | 0.0215 |
| | trojan_wm | 0.0220 | 0.9089 | 0.7435 | 0.7513 | FT-AL | 0.8684 | 0.3257 | 0.1726 | 0.0117 |
| | | | | | | FT-LL | 0.5935 | 0.7429 | 0.5751 | 7.4281e-11 |
| | | | | | | RT-AL | 0.8519 | 0.1170 | 0.0684 | 0.0000 |
| | | | | | | Pruning-20% | 0.8647 | 0.3235 | 0.1779 | 0.0131 |
| | | | | | | Pruning-50% | 0.8610 | 0.3281 | 0.1747 | 0.0000 |

Table 2: Evaluation of watermarking against fine-tuning and pruning on three datasets.

## 4.2 DEFENDING AGAINST FINE-TUNING & PRUNING

We evaluate the robustness of our proposed method against fine-tuning and pruning in Table 2, where victim models are watermarked models, and suspect models are stolen copies of victim models using watermark removal attacks. OoDWSR of the pre-trained model in Table 1 is the probability that a non-watermarked model classifies the verification samples as the target label. If the OoDWSR of a suspect model far exceeds that of the non-watermarked model, the suspect model can be justified as a copy of the victim model (Jia et al., 2021).

FT-AL and pruning maintain the performance of the main classification task with an accuracy degradation of less than $6\%$, but OoDWSR remains high for all the datasets. Compared with FT-AL, FT-LL will significantly bring down the standard accuracy by over $15\%$ for all the datasets. Even with the large sacrifice of standard accuracy, FT-LL still cannot wash out the injected watermark, and the OoDWSR even increases for some of the datasets. RT-AL loses $4.50\%$, $16.63\%$, and $5.47\%$ (mean value for two triggers) standard accuracy respectively for three datasets. Yet, OoDWSR in RT-AL is larger than the one of the random guess and non-watermarked models. To statistically verify the ownership, we conduct a T-test between the non-watermarked model and the watermarked model. The p-value is the probability that the two models behave similarly. p-values for all the datasets are close to $0$. The low $p$-values indicate that the suspect models have significantly different behaviors compared with non-watermarked models in probability, at least $95\%$. Thus, these suspect models cannot get rid of the suspicion of copying our model $\mathcal{M}$ with a high chance.

IDWSR is also used here as a reference, although we do not use i.i.d. data for verification of the ownership of our model. We observe that even though watermark can be successfully injected into

| Trigger | Training data | Victim model | | | Suspect model | | |
|---|---|---|---|---|---|---|---|
| | | Acc | IDWSR | OoDWSR | Acc | IDWSR | OoDWSR |
| trojan_wm | clean | 0.9400 | 0.0639 | 0.0487 | 0.8646 | 0.0864 | 0.0741 |
| | ID | 0.9378 | 1.0000 | 0.9997 | 0.8593 | 0.0413 | 0.0195 |
| | OoD | 0.9102 | 0.9768 | 0.9566 | 0.8706 | 0.4434 | **0.5752** |
| trojan_8x8 | clean | 0.9400 | 0.0161 | 0.0481 | 0.8646 | 0.0323 | 0.0610 |
| | ID | 0.9393 | 0.9963 | 0.9992 | 0.8598 | 0.0342 | 0.0625 |
| | OoD | 0.9178 | 0.9328 | 0.9423 | 0.8675 | 0.0782 | **0.2419** |

Table 3: Comparison of watermarking methods against fine-tuning watermark removal using different training data. OoD injection is much more robust compared with i.i.d. injection.

both our generated OoD dataset and i.i.d. samples (refer to IDWSR and OoDWSR for victim model), they differ in their robustness against these two watermark removal attacks. For instance, for smooth of GTSRB, after fine-tuning or pruning, IDWSR drops under $1\%$, which is below the random guess, however, OoDWSR remains over $67\%$. This phenomenon is also observed for other triggers and datasets. Watermarks injected in OoD samples are much harder to be washed out compared with watermarks injected into i.i.d. samples. Due to different distributions, fine-tuning or pruning will have a smaller impact on OoD samples compared with i.i.d. samples.

To further verify our intuition, we also compare our method (OoD) with traditional backdoor-based methods using i.i.d. data (ID) for data poisoning on CIFAR-10. We use RT-AL which is the strongest attack in Table 2 as an example. The results are shown in Table 3. Note that ID poison and the proposed OoD poison adopt IDWSR and OoDWSR as the success rate for the injection watermark, respectively. Clean refers to the pre-trained model without watermark injection. With only one single OoD image for watermark injection, we can achieve comparable results as ID poisoning which utilizes the entire ID training set. After RT-AL, the watermark success rate drops to $4.13\%$ and $3.42\%$, respectively for ID poison, while drops to $57.52\%$ and $24.19\%$ for OoD poison, which verifies that our proposed method is also much more robust against watermark removal attacks.

| Dataset | Trigger | Victim model | | | Suspect model | | | p-value |
|---|---|---|---|---|---|---|---|---|
| | | Acc | IDWSR | OoDWSR | Acc | IDWSR | OoDWSR | |
| CIFAR-10 | trojan_wm | 0.9102 | 0.9768 | 0.9566 | 0.8485 | 0.9684 | 0.9547 | 0.0000 |
| | trojan_8x8 | 0.9178 | 0.9328 | 0.9423 | 0.8529 | 0.8882 | 0.9051 | 0.0000 |
| CIFAR-100 | trojan_8x8 | 0.6978 | 0.7024 | 0.8761 | 0.5309 | 0.5977 | 0.7040 | 0.0059 |
| | l0_inv | 0.6948 | 0.7046 | 0.5834 | 0.5200 | 0.0162 | 0.0622 | 0.0019 |
| GTSRB | smooth | 0.9146 | 0.1329 | 0.9442 | 0.6575 | 0.1386 | 0.9419 | 7.5891e-11 |
| | trojan_wm | 0.9089 | 0.7435 | 0.7513 | 0.6379 | 0.7298 | 0.7666 | 2.6070e-21 |

Table 4: Evaluation of watermarking against model extraction watermark removal on three datasets.

### 4.3 DEFENDING AGAINST MODEL EXTRACTION

We evaluate the robustness of our proposed method against model extraction in Table 4. By conducting model extraction, the standard accuracy drops $6\%$ on the model pre-trained on CIFAR-10, and drops more than $10\%$ on the other two datasets. Re-training from scratch makes it hard for the suspect model to resume the original model's utility using an OoD dataset and soft labels querying from the watermarked model. OoDWSR is still over $90\%$ and $76\%$ for CIFAR-10 and GTSRB, respectively. Although OoDWSR is $6.22\%$ for l0_inv, it is still well above $0.02\%$, which is observed for the non-watermarked model. All the datasets also have a p-value close to 0. All the above observations indicate that the re-training-based extracted model has a high probability of being a copy of our model. One possible reason for these re-training models still extracting the watermark is that during re-training, the backdoor information hidden in the soft label queried by the IP infringers can also embed the watermark in the extracted model. The extracted model will behave more similarly to the victim model as its decision boundary gradually approaches that of the victim model.

### 4.4 QUALITATIVE STUDIES

**Distribution of generated OoD samples and ID samples.** We first augment an unlabeled OoD dataset, and then assign predicted labels to them using the model pre-trained on clean CIFAR-10 data. According to the distribution of OoD and ID samples before and after our watermark fine-tuning as shown in Fig. 3, we can observe that the OoD data drawn from one image lies close to ID data with a small gap. After a few epochs of fine-tuning, some of the OoD data is drawn closer to ID,

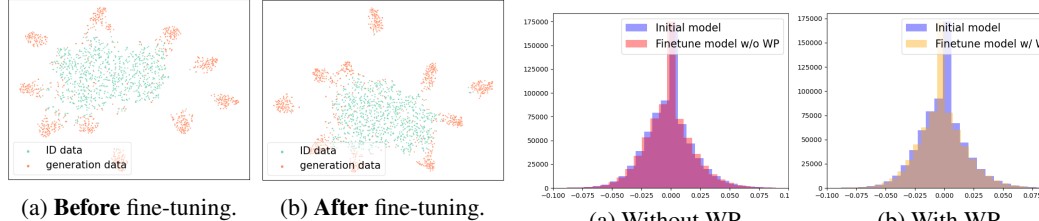

(a) **Before** fine-tuning.  (b) **After** fine-tuning.

Figure 3: The distribution of OoD and ID samples. Generation data denotes augmented OoD samples from a single OoD image.

(a) Without WP.  (b) With WP.

Figure 4: Weight distribution for model w/ and w/o WP. The x-axis is the parameter values, and the y-axis is the number of parameters.

| OoD Image | Trigger | Acc | IDWSR | OoDWSR |
|---|---|---|---|---|
| City | trojan_wm | 0.9102 | 0.9768 | 0.9566 |
| | trojan_8x8 | 0.9178 | 0.9328 | 0.9423 |
| Animals | trojan_wm | 0.9072 | 0.9873 | **0.9880** |
| | trojan_8x8 | 0.9176 | 0.9251 | 0.9622 |
| Bridge | trojan_wm | 0.9207 | 0.8749 | 0.7148 |
| | trojan_8x8 | 0.9172 | 0.7144 | 0.7147 |

Table 5: Watermark injection using different OoD images.

| Trigger | WP | Victim model | | | Suspect model | | |
|---|---|---|---|---|---|---|---|
| | | Acc | IDWSR | OoDWSR | Acc | IDWSR | OoDWSR |
| trojan_wm | w/o | 0.9264 | 0.9401 | 0.9490 | 0.8673 | 0.1237 | 0.1994 |
| | w/ | 0.9102 | 0.9768 | 0.9566 | 0.8706 | 0.4434 | **0.5752** |
| trojan_8x8 | w/o | 0.9238 | 0.9263 | 0.9486 | 0.8690 | 0.0497 | 0.1281 |
| | w/ | 0.9178 | 0.9328 | 0.9423 | 0.8675 | 0.0782 | **0.2419** |

Table 6: Weight perturbation increases the robustness of the watermarks against removal attacks.

but still maintains no overlap. This can help us successfully implant watermarks to the pre-trained model while maintaining the difference between ID and OoD data. In this way, when our model is fine-tuned with clean ID data by attackers, the WSR on the OoD data will not be easily erased.

**Effects of different OoD images for watermark injection.** In Table 5, we use different source images to generate surrogate datasets and inject watermarks into a pre-trained model. The model is pre-trained on CIFAR-10. From these results, we observe that the choice of the OoD image for injection is also important. Dense images such as "City" and "Animals" can produce higher OoDWSR than the sparse image "Bridge", since more knowledge is included in the visual representations of dense source images. Thus, dense images perform better for backdoor-based watermark injection. This observation is also consistent with some previous arts (Asano & Saeed, 2023; Asano et al., 2019) about single image representations, which found that dense images perform better for model distillation or self-supervised learning.

**Effects of backdoor weight perturbation.** We show the results in Fig. 4. The initial model is WideResNet pre-trained on CIFAR-10, and the fine-tuned model is the model fine-tuning using our proposed method. If the OoD data is directly utilized to fine-tune the pre-trained models with only a few epochs, the weight distribution is almost identical for pre-trained and fine-tuned models (left figure). According to Garg et al. (2020), if the parameter perturbations are small, the backdoor-based watermark can be easily removed by fine-tuning or adding random noise to the model's parameters. Our proposed watermark injection WP (right figure) can shift the fine-tuned model parameters from the pre-trained models in a reasonable scale compared with the left one, while still maintaining high standard accuracy and watermark success rate as shown in Table 6. Besides, the weight distribution of the perturbed model still follows a normal distribution as the unperturbed model, performing statistical analysis over the model parameters distributions will not be able to erase our watermark.

To show the effects of WP, we conduct the attack RT-AL on CIFAR-10 as an example. From Table 6, we observe that WP does not affect the model utility, and at the same time, it will become more robust against stealing threats, since OoDWSR increases from 19.94% and 12.81% to 57.52% and 24.19%, respectively, for two triggers. More results for WP can be referred to Appendix A.2.

## 5 CONCLUSION

In this paper, we proposed a novel and practical watermark injection method that does not require training data and utilizes a single out-of-distribution image in a sample-efficient and time-efficient manner. We designed a robust weight perturbation method to defend against watermark removal attacks. Our extensive experiments on three benchmarks showed that our method efficiently injected watermarks and was robust against three watermark removal threats. Our approach has various real-world applications, such as protecting purchased models by encoding verifiable identity and implanting server-side watermarks in distributed learning when ID data is not available.

ACKNOWLEDGEMENT

This material is based in part upon work supported by the National Science Foundation under Grant IIS-2212174, IIS-1749940, Office of Naval Research N00014-20-1-2382, N00014-24-1-2168, and National Institute on Aging (NIA) RF1AG072449. The work of Z. Wang is in part supported by the National Science Foundation under Grant IIS2212176.

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
