## A    METHODOLOGY SUPPLEMENTARIES

### A.1    EXTENDED WATERMARK INJECTION RESULTS

**Additional watermark injection results.** Fig. 5 shows the watermark injection for CIFAR-100 and GTSRB. It only takes 20 epochs for CIFAR-100 and GTSRB to achieve stable high standard accuracy and OoDWSR. The highest OoDWSR for CIFAR-100, and GTSRB are $0.8761$, and $0.9442$, respectively, with standard accuracy degradation of less than $3\%$.

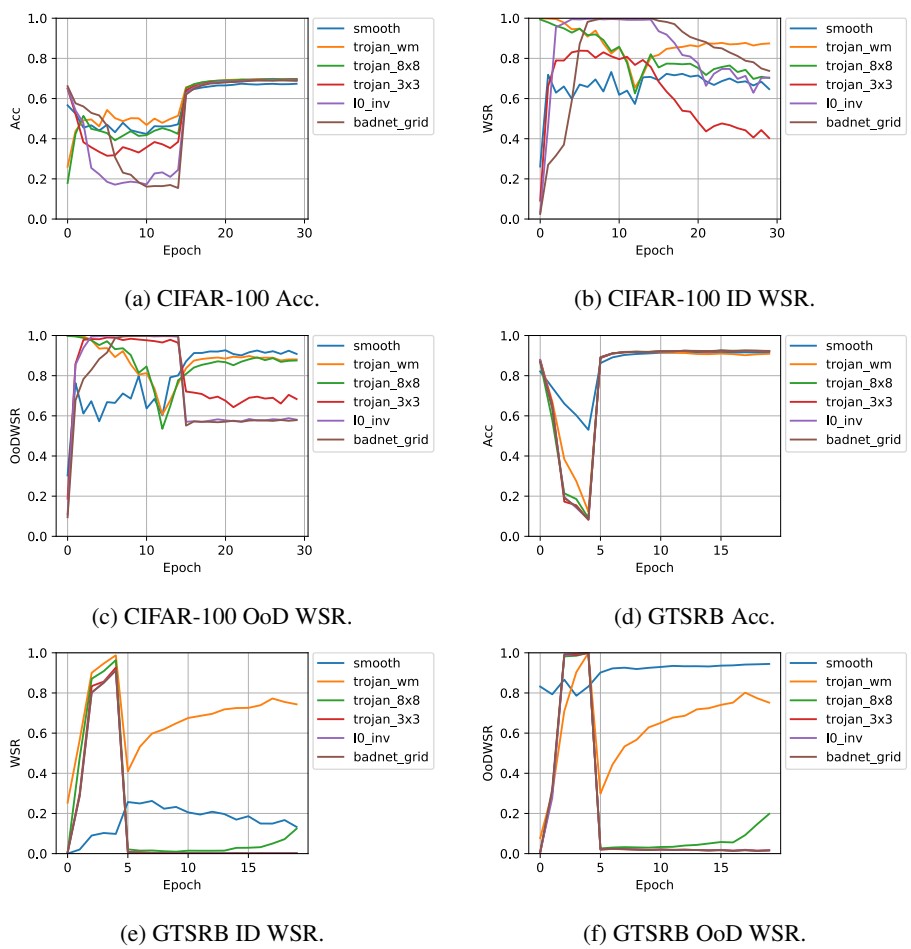

(a) CIFAR-100 Acc.                    (b) CIFAR-100 ID WSR.

(c) CIFAR-100 OoD WSR.                    (d) GTSRB Acc.

(e) GTSRB ID WSR.                    (f) GTSRB OoD WSR.

Figure 5: Acc, ID WSR, and OoD WSR for watermark injection. The watermarks are injected quickly with high accuracy and OoDWSR. Triggers with the highest OoDWSR and accuracy degradation of less than $3\%$ are selected for each dataset.

### A.2    EXTENDED WEIGHT PERTURBATION RESULTS

**Additional experiments about the effects of WP.** Table 7 shows the results for fine-tuning method FT-AL. We observe that by applying WP, after fine-tuning, we can increase OoDWSR from $0.7305$ and $0.8184$ to $0.9678$ and $0.9797$, respectively.

To further verify the robustness of our proposed weight perturbation, in Fig. 6, we also show the results of a much more challenging setting, i.e., RT-AL with $100\%$ training data. With more data for fine-tuning, RT-AL can obtain an average standard accuracy of $0.9069$ and $0.9074$, respectively for the model w/ and w/o WP. With comparable standard accuracy, WP can increase OoDWSR by $55.45\%$, $0.35\%$, $5.02\%$, $16.23\%$ for trojan_wm, trojan_8x8, trojan_3x3, l0_inv, respectively. With WP, during the fine-tuning process, OoDWSR will remain more stable or even increase with the

| Trigger | WP | Victim model | | | Suspect model | | |
|---------|-----|------|-------|--------|------|--------|--------|
| | | Acc | IDWSR | OoDWSR | Acc | IDWSR | OoDWSR |
| trojan_wm | w/o | 0.9264 | 0.9401 | 0.9490 | 0.9226 | 0.6327 | 0.7305 |
| | w/ | 0.9102 | 0.9768 | 0.9566 | 0.9191 | 0.9769 | **0.9678** |
| trojan_8x8 | w/o | 0.9238 | 0.9263 | 0.9486 | 0.9223 | 0.5304 | 0.8184 |
| | w/ | 0.9178 | 0.9328 | 0.9423 | 0.9187 | 0.9533 | **0.9797** |

Table 7: Weight perturbation increases the robustness of the watermarks against removal attacks (FT-AL).

increase of standard accuracy. These results demonstrate that the proposed WP can help our injected watermark be more robust and persistent even under more challenging stealing threats.

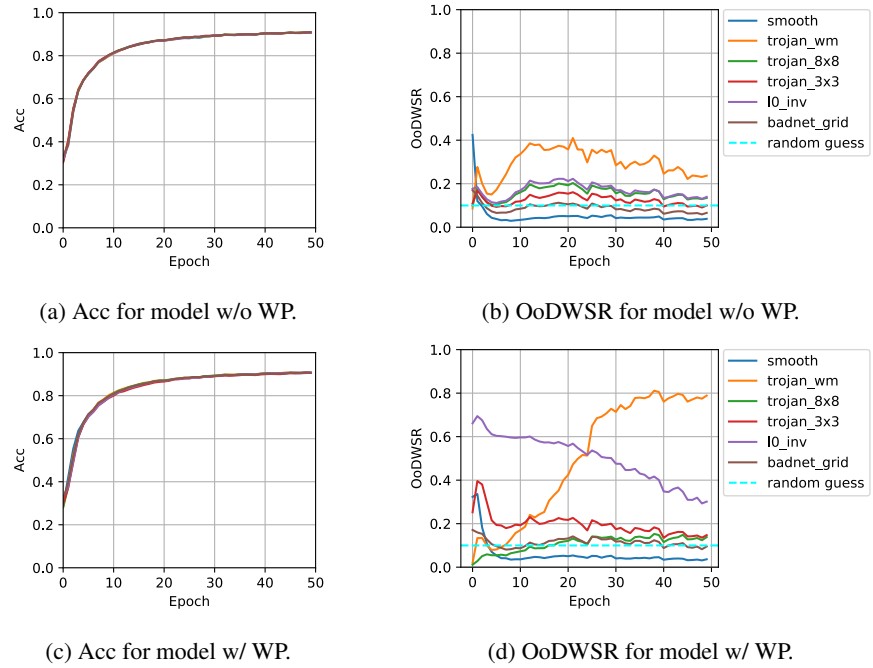

(a) Acc for model w/o WP.  (b) OoDWSR for model w/o WP.

(c) Acc for model w/ WP.  (d) OoDWSR for model w/ WP.

Figure 6: Effects of weight perturbation against RT-AL with $100\%$ training data on CIFAR-10.

**Watermark loss landscape for WP.** To get the watermark loss landscape around the watermarked model $\mathcal{M}_w(\theta_w)$, we interpolate between the initial pre-trained model $\mathcal{M}(\theta)$ and the watermarked model $\mathcal{M}_w(\theta_w)$ along the segment $\theta_i = (1-t)\theta + t\theta_w$, where $t \in [0,1]$ with increments of 0.01. The watermark loss is evaluated on the verification dataset composed of the generated OoD samples. Pre-trained model $\mathcal{M}(\theta)$ and watermarked model $\mathcal{M}_w(\theta_w)$ correspond to coefficients of 0 and 1, respectively. From Fig. 7, we observe that with WP, we can achieve a flatter loss landscape (orange line) around the point of the watermarked model compared with the one without WP (blue line). By maximizing the backdoor loss in Eq. (3) over the perturbation $v$ we can get a flatter landscape of watermark loss around the optimal point of the watermarked model. For those watermark removal attacks such as fine-tuning or pruning, which might make a minor parameter change to the models, a flatter loss landscape could prevent the model from escaping from the watermarked local optimum compared with sharp ones. The loss landscape of trojan_wm is flatter than trojan_8x8, thus, the robustness of trojan_wm is also better than trojan_8x8 as shown in Table 2 in our paper.

To show the relationship between robustness of our proposed method against watermark removal attacks and the loss landscape, we also interpolate between the suspect model after clean data fine-tuning $\mathcal{M}_s(\theta_s)$ and the watermarked model $\mathcal{M}_w(\theta_w)$ along the segment $\theta_i = (1-t)\theta_s + t\theta_w$, where $t \in [0,1]$ with increments of 0.01. Suspect model $\mathcal{M}_s(\theta_s)$ and watermarked model $\mathcal{M}_w(\theta_w)$ correspond to coefficients of 0 and 1, respectively. Fig. 8 shows the loss for the interpolated model between the suspect model after clean i.i.d. data fine-tuning (FT-AL and RT-AL) and our watermarked model. We observe that the flatness of the loss landscape can lead to a lower watermark loss during

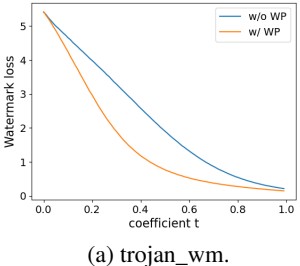
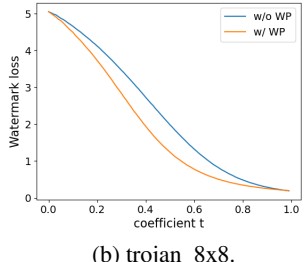

(a) trojan_wm.  (b) trojan_8x8.

Figure 7: Watermark loss landscape from pre-trained model to watermarked model on CIFAR-10.

fine-tuning, which makes it harder for IP infringers to escape from the local optimum. These results combinedly give an explanation for why our proposed WP can increase the robustness of the watermark against watermark removal attacks.

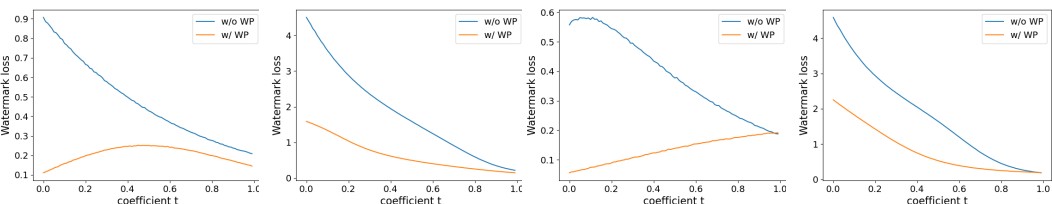

(a) FT-AL for trojan_wm.  (b) RT-AL for trojan_wm.  (c) FT-AL for trojan_8x8.  (d) RT-AL for trojan_8x8.

Figure 8: Watermark loss landscape from suspect model to watermarked model on CIFAR-10.

### A.3 EXTENDED ROBUSTNESS ANALYSIS

**Defending against OoD detection.** Kim et al. (2023) found that the adversary may simply reject the query of OoD samples to reject the ownership verification process using the energy-based out-of-distribution detection method (Liu et al., 2020). We compute the mean energy score for ID samples and our verification samples, and also report the area under the receiver operating characteristic curve (AUROC), and the area under the PR curve (AUPR) for ID and OoD classification according to (Liu et al., 2020) in Table 8. For the watermarked model pre-trained on CIFAR-100 and GTSRB, the energy scores for ID and verification samples are very close, and the AUROC and AUPR are close to random guesses. For CIFAR-10, despite the differences of energy score, AUPR for OoD detection is only 20%, which indicates that it will be very hard for the adversaries to filter out verification samples from ID ones, if verification samples are mixed with other samples while querying the suspect model. Therefore, the query of our verification samples cannot be found out by the IP infringers. A possible reason is that the distribution of our generated verification samples lies closely to the ID ones, since previous work (Asano & Saeed, 2023) shows that the generated samples from the single OoD image can also be used to train a classifier yielding reasonable performance on the main prediction task.

| Dataset | Trigger | Energy of ID samples | Energy of verification samples | AUROC | AUPR |
|---------|---------|---------------------|-------------------------------|-------|------|
| CIFAR-10 | trojan_wm | -5.2880 | -14.0207 | 0.8160 | 0.2068 |
| | trojan_8x8 | -5.1726 | -11.2574 | 0.8107 | 0.2028 |
| CIFAR-100 | trojan_8x8 | -14.3086 | -13.4729 | 0.4487 | 0.0757 |
| | l0_inv | -13.9304 | -11.9693 | 0.3614 | 0.0651 |
| GTSRB | smooth | -8.6203 | -8.2872 | 0.4564 | 0.0715 |
| | trojan_wm | -8.3042 | -8.4407 | 0.4922 | 0.0646 |

Table 8: Energy-based OoD detection between ID and verification samples. Higher AUROC and AUPR indicate better OoD detection performance.

A.4   EXTENDED ABLATION SUDUDIES

**Effects of different numbers of verification samples.** We generate 45000 surrogate samples for watermark injection, but we do not need to use all the surrogate data for verification. We report the OoDWSR w.r.t. different numbers of samples in the verification dataset in Fig. 9. According to the figure, only 450 verification samples are enough for accurate verification.

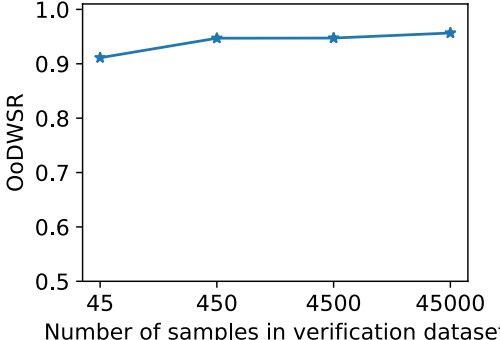

Figure 9: OoDWSR w.r.t. different numbers of samples in the verification dataset. The model is pre-trained on CIFAR-10, and the trigger pattern is trojan_wm.

**Effects of different number of source OoD images.** We evaluate the effect of a combination of

| Source OoD Image | Victim model | | | Suspect model | | | p-value |
|---|---|---|---|---|---|---|---|
| | Acc | IDWSR | OoDWSR | Acc | IDWSR | OoDWSR | |
| City | 0.9102 | 0.9768 | 0.9566 | 0.9191 | 0.9769 | 0.9678 | 0.0000 |
| Animal | 0.9072 | 0.9873 | 0.9880 | 0.9212 | 0.8309 | 0.8922 | 0.0000 |
| City+Animal | 0.9059 | 0.9820 | 0.9638 | 0.9209 | 0.8311 | 0.8511 | 0.0000 |

Table 9: Effects of different number of source OoD images.

different source OoD images in Table 9. The model is pre-trained on CIFAR-10, and trojan_wm is adopted as the trigger pattern. The Suspect model is fine-tuned using FT-AL by the IP infringer. According to the results, for watermark injection (see victim model), with a combination of two OoD images, OoDWSR is between the values for the two combination images. For the robustness against fine-tuning conducted by the IP infringer (see suspect model), OoDWSR even decreases after the combination of the source OoD images. However, one advantage of using multiple OoD images is that it will be harder for the IP infringers to get access to the source OoD image and guess the composition of our validation set, which can further enhance the security of the watermark.