# OpenReview forum: "Safe and Robust Watermark Injection with a Single OoD Image"
_ICLR.cc/2024/Conference — ICLR 2024 poster_

### Official Review · Reviewer_suCZ · 2023-10-24

**Soundness:** 2 fair
**Presentation:** 3 good
**Contribution:** 2 fair
**Rating:** 6
**Confidence:** 4

**Summary:**

The submission proposes a backdoor-based model watermarking scheme with a single OOD image. The work approaches a new scenario that the verifier of the model has no access to the original training data (e.g., as a third party or model-sharing platform), and demonstrates the robustness and efficiency of the watermark injection method.

**Strengths:**

+ The submission focuses on a new task that the watermark injection is performed after the training phrase, which allows the IP protection in more real-world scenarios.
+ The proposed watermark injection method only requires a single OOD image to insert the copyright information, reducing the cost of sampling trigger datasets whilst maintaining the confidentiality.

**Weaknesses:**

+ **About the *Model Extraction* attack**: It is mentioned in section 4 that *the behavior of a victim model is cloned by fine-tuning it with the queried image-prediction pairs*, then what's the difference between the fine-tuning attack and model extraction? As retraining-based extraction attacks are generally considered in previous works [1-2], the robustness against such an attack should be further demonstrated.
+ **Possible OOD detection**: As the verification process is conducted with the augmentation of a single OOD image, the adversary may simply reject the query of OOD samples to reject the ownership verification process, as illustrated in [2]. Since OOD detection is entirely possible in the realistic scenarios proposed by the submission, the adpative defenses shoule be discussed in further detail.

References:

[1] Jia H, Choquette-Choo C A, Chandrasekaran V, et al. Entangled watermarks as a defense against model extraction. 30th USENIX Security Symposium (USENIX Security 21). 2021: 1937-1954.

[2] Kim B, Lee S, Lee S, et al. Margin-based Neural Network Watermarking. Proceedings of the 40th International Conference on Machine Learning. 2023.

**Questions:**

As the submission highlights the use of a single OOD image for watermark injection, the author may provide more comparison between the "pachify" of a single image and sampling several OOD images as the verification dataset, as to enhance the persuasiveness. Besides, more adaptive defenses such as OOD detection and Neural Cleanse [3] should be considered in addition to the traditional watermark removal attacks.

Reference:

[3] Wang B, Yao Y, Shan S, et al. Neural cleanse: Identifying and mitigating backdoor attacks in neural networks. 2019 IEEE Symposium on Security and Privacy (SP). IEEE, 2019: 707-723.

---

> ### Author Response · Authors · 2023-11-17
> **Thank you for your helpful comment and suggestions**
>
> We thank the reviewer for the constructive comments and suggestions, which we address below:
> 1) **About the Model Extraction attack: It is mentioned in section 4 that the behavior of a victim model is cloned by fine-tuning it with the queried image-prediction pairs, then what's the difference between the fine-tuning attack and model extraction? As retraining-based extraction attacks are generally considered in previous works [1-2], the robustness against such an attack should be further demonstrated.**
>
> Thank you very much for your comment. We add the results for re-training-based model extraction in Table 8 in the supplementary. For this method, a large standard accuracy degradation (more than 10%) is observed, but the watermark is still extracted by the suspect model with high OoDWSR and p-value close to 0. We have included a detailed analysis of the results in the supplementary.
>
> 2) **Possible OOD detection: As the verification process is conducted with the augmentation of a single OOD image, the adversary may simply reject the query of OOD samples to reject the ownership verification process, as illustrated in [2]. Since OOD detection is entirely possible in the realistic scenarios proposed by the submission, the adpative defenses shoule be discussed in further detail.**
>
> Thank you very much for your suggestion.  Following [2], we add one experiment using energy score [3] for OoD detection in Table 9 in the supplementary.  We compute the mean energy score for ID samples and our verification samples, and also AUROC and AUPR for ID and OoD classification according to [3]. AUROC and AUPR are two frequently used evaluation metrics for OoD detection. The higher these two metrics, the better the OoD detection performance. For the watermarked model pre-trained on CIFAR-100 and GTSRB, the energy scores for ID and verification samples are very close, and the AUROC and AUPR are close to random guesses. For CIFAR-10, despite the differences of energy score, AUPR for OoD detection is only 20\%, which indicates that it will be very hard for the adversaries to filter out verification samples from ID ones, if verification samples are mixed with other samples while querying the suspect model. Therefore, the query of our verification samples cannot be found out by the IP infringers. A possible reason is that the distribution of our generated verification samples lies closely to the ID ones, since previous work [4] shows that the generated samples from the single OoD image can also be used to train a classifier yielding reasonable performance on the main prediction task.
>
> 3) **As the submission highlights the use of a single OOD image for watermark injection, the author may provide more comparison between the "pachify" of a single image and sampling several OOD images as the verification dataset, as to enhance the persuasiveness. Besides, more adaptive defenses such as OOD detection and Neural Cleanse should be considered in addition to the traditional watermark removal attacks.**
>
> Thank you very much for your suggestions. We investigate the effect of different OoD images in Table 5, and the combination of different OoD images in Table 10 in the supplementary. We also add the results of OoD detection in Table 9 in the supplementary as explained in the last question.
>
> [1] Jia H, Choquette-Choo C A, Chandrasekaran V, et al. Entangled watermarks as a defense against model extraction. 30th USENIX Security Symposium (USENIX Security 21). 2021: 1937-1954.
>
> [2] Kim B, Lee S, Lee S, et al. Margin-based Neural Network Watermarking. Proceedings of the 40th International Conference on Machine Learning. 2023.
>
> [3] Weitang Liu, Xiaoyun Wang, John Owens, and Yixuan Li. Energy-based out-of-distribution detection. Advances in neural information processing systems, 33:21464–21475, 2020
>
> [4] Yuki M. Asano and Aaqib Saeed. Extrapolating from a single image to a thousand classes using distillation. In ICLR, 2023

---

> ### Author Response · Authors · 2023-11-21
> **A kind reminder to reviewer suCZ**
>
> Dear Reviewer suCZ,
>
> Thank you for your time to review our paper and leave valuable comments and suggestions. We are wondering whether you have got a chance to read our response to your questions. We will be glad to provide more explanations and answer more questions if you have any.
>
> Authors

---

### Official Review · Reviewer_URnu · 2023-10-29

**Soundness:** 3 good
**Presentation:** 3 good
**Contribution:** 3 good
**Rating:** 8
**Confidence:** 4

**Summary:**

This paper proposes a novel technique for protecting the intellectual property of deep models using a single out-of-distribution image as a secret key. The proposed technique is safe and robust against watermark removal attacks and does not require poisoning training samples.

**Strengths:**

The main strength of this paper is the proposal of a novel technique for protecting the intellectual property of deep models using a single out-of-distribution (OoD) image as a secret key. The proposed technique is robust against watermark removal attacks and does not require poisoning training samples. The paper has great clarity and is easy to follow. The proposed technique looks promising for protecting the commercial ownership of deep models, which require large amounts of data and computational resources to train.

The main technical difference between this work and prior work is the use of a single OoD image as a secret key for watermark injection. Prior work on watermarking strategies often implants verifiable backdoor triggers by poisoning training samples, which are often unrealistic due to data privacy and safety concerns and are vulnerable to minor model changes such as fine-tuning. In contrast, the proposed technique leverages the diverse knowledge from a single OoD image to inject backdoor-based watermarks efficiently to different parts of the pre-trained representation space. Additionally, the proposed technique is safe and robust against watermark removal attacks and does not require poisoning training samples. Since the training data is independent, it is also not necessary to trust third-party providers with sensitive data.

Experimental findings demonstrate that the proposed watermarking approach is effective in protecting IP of deep models. The authors show that their technique is robust against common watermark removal attacks, including fine-tuning, pruning, and model extraction. The proposed technique is also time- and sample-efficient without requiring any training data. The authors demonstrate the effectiveness of their approach on several popular models including VGG16, ResNet50, and MobileNetV2. The experimental results show that the proposed technique achieves high watermark detection accuracy while maintaining the performance of the original model on the original task.

**Weaknesses:**

Relying on a single OoD image as a secret key might introduce vulnerabilities. If an adversary gains access to this single image, the watermarking scheme could be compromised. Using multiple OoD images or a combination of techniques might enhance security?

 It is unclear why specific trigger patterns are well suited for specific dataset. Further, it would be interesting to see how the watermarking technique performs when models are transferred across different tasks or domains. The watermark might lose its effectiveness in such scenarios.

**Questions:**

How many verification samples are needed for each successful verification?

In experiments the authors tried many different triggers. which trigger should an IP protector choose in practice?

---

> ### Author Response · Authors · 2023-11-17
> **Thank you for your helpful comment and suggestions**
>
> We are glad that the reviewer found our approach novel, robust, and effective. We thank the reviewer for the constructive comments and suggestions, which we address below:
> 1) **Relying on a single OoD image as a secret key might introduce vulnerabilities. If an adversary gains access to this single image, the watermarking scheme could be compromised. Using multiple OoD images or a combination of techniques might enhance security?**
>
> Thank you very much for your suggestion. We add the evaluation for the effect of a combination of different source OoD images in Table 10 in the supplementary. Although the watermark is not more robust against fine-tuning conducted by the IP infringer, we agree with your point that using multiple OoD images will make it harder for the IP infringers to get access to the source OoD image and guess the composition of our validation set, which can further enhance the security of the watermark.
>
> 2) **It is unclear why specific trigger patterns are well suited for specific dataset. Further, it would be interesting to see how the watermarking technique performs when models are transferred across different tasks or domains. The watermark might lose its effectiveness in such scenarios.**
>
> An interesting scenario you have mentioned! We think re-training-based model extraction attacks using data from different domains as the auxiliary dataset might be similar to the scenario you have mentioned. It retrains a student model using the auxiliary data and the label queried from the victim (teacher) model. We add the results and corresponding analysis for this method in Table 8 in the supplementary. The results show that our method is still robust against this extraction attack using data from different domains.
>
> 3) **How many verification samples are needed for each successful verification?**
>
> We investigate the effects of different numbers of verification samples in Figure 9 in the supplementary. We generate 45000 surrogate samples for watermark injection, but we do not need to use all the surrogate data for verification. We report the OoDWSR w.r.t. different numbers of samples in the verification dataset in Fig. 9. According to the figure, only 450 verification samples are enough for accurate verification.
>
> 4) **In experiments the authors tried many different triggers. which trigger should an IP protector choose in practice?**
>
> Thank you very much for your comment. Trigger patterns in this paper are derived from prior arts, of which no trigger could be claimed to be successfully injected on all datasets with 100\% WSR. Whether these triggers can be injected successfully is related to the pattern instead of our proposed method. In practice, we can search for a valid pattern from a large enough candidate set that covers as many existing trigger patterns as possible. As we have done in our experiments, we found proper triggers for all evaluated datasets.

---

> > ### Comment · Reviewer_URnu · 2023-11-20
> > **Thanks for response**
> >
> > I appreciate the authors' response, my concerns about this paper have been well addressed.

---

> > > ### Author Response · Authors · 2023-11-21
> > >
> > > Thank you very much for carefully reading our response! We are glad our response has addressed your concerns.

---

### Official Review · Reviewer_MV1f · 2023-10-31

**Soundness:** 2 fair
**Presentation:** 2 fair
**Contribution:** 2 fair
**Rating:** 3
**Confidence:** 4

**Summary:**

This paper presents a new method to inject watermarks into the DNN models by leveraging only one out-of-distribution image. It first crops some patches from the selected image, augments, and adds trigger patterns on some of them as the trigger images. The rest of the patches are also augmented but not injected with the trigger patterns, and used as the data to retain the main functionality of the target model. Then, it jointly fine-tunes the target model using the two parts of augmented data. Meanwhile, in order to improve the watermark robustness under the watermark removal attacks, the authors propose to randomly add perturbations to the model weights during fine-tuning. The authors claim they can achieve a better performance compared with that implemented using basic backdoor-based methods.

**Strengths:**

The proposed approach is interesting, and the paper is easy to follow.

**Weaknesses:**

1. The authors do not give a comprehensive discussion of previous work on this topic.

2. The experimental justification of this work is not sufficient, only compared to the basic backdoor-based strategy.

**Questions:**

The questions are listed based on the organization order of the paper:

1. On page 2. ‘…, which fills in the gap of IP protection of deep models without training data’. Most of the white-box watermarking methods do not need the training data, you should make it clear that you target the backdoor-based methods.

2. On page 5. ‘If the protected model shares a similar parameter distribution with the pre-trained model, the injected watermark could be easily erased by fine-tuning using clean i.i.d. data or adding random noise to parameters’. What is the pre-trained model? Does it mean that fine-tuning with the i.i.d data can introduce more perturbations on the watermarks than fine-tuning with OoD data? What’s the intuition behind such an claim?

3. On page 5. The statement of ‘the anomalies of the parameter distribution could be easily detected by an IP infringer’ may be conflict with the experimental results of Ref. [1]. The authors of Ref. [1] conduct experiments to check whether the distributions of model weights or model representations could be used as indicators to detect backdoors or anomalies in a DNN model.

4. Experimental settings are not clear enough: 1) How many patches are used for training? 2) Comparison experiments are not well organized. There are many previous methods with similar strategies, e.g., BadNets, 10-invisible, smooth, Trojan Square, and Trojan watermark, and it's essential to compare with them.

5. In Table 2, we observe that the OoDWSR may increase after fine-tuning and pruning. Could you give some explanations? It seems the IDWSR may also increase after such two attacks. Why? It seems to be contradicted by your previous claim.

6. How did you implement the model extraction attack? According to the experimental results in Table 4, we still have a high OoDWSR after model extraction. Could you explain why the model extraction methods using i.i.d data can also extract the watermarks that are generated using the OoD data?

[1] Ji, Y., Liu, Z., Hu, X., Wang, P., & Zhang, Y. (2019). Programmable neural network trojan for pre-trained feature extractor. *arXiv preprint arXiv:1901.07766*.

---

> ### Author Response · Authors · 2023-11-17
> **Thank you for your helpful comment and suggestions - Part 1**
>
> We are glad that the reviewer found our approach interesting. We thank the reviewer for the constructive comments and suggestions, which we address below:
> 1) **The authors do not give a comprehensive discussion of previous work on this topic.**
>
> Thank you very much for your comment. We discuss the DNN watermarking in section 2.1, including parameter-based watermarking, backdoor-based watermarking, and a special kind of backdoor-based watermarking without i.i.d data, which is closely related to our problem setting. We also discuss watermark removal attack in section 2.2.
>
> 2) **The experimental justification of this work is not sufficient, only compared to the basic backdoor-based strategy.**
>
> Thank you very much for your comment. We want to argue that the problem setting for our proposed method and existing backdoor-based strategies are different. The existing backdoor-based strategy requires training data or in-distribution (ID) data to maintain the model utility, while our paper explores a more realistic scenario where access to ID data can be prohibitive due to data safety and confidentiality.
>
> Despite the different settings, we also compare with traditional methods to further verify the robustness of our method. The traditional backdoor-based method uses in-distribution (ID) data to inject watermarks, we compare our method with the traditional backdoor-based method in Table 3. According to the results, after RT-AL, the watermark success rate drops to 4.13% and 3.42%, respectively for ID poison (basic backdoor), while drops to 57.52% and 24.19% for OoD poison (proposed method), which verifies that our proposed method is also much more robust against watermark removal attacks.
>
> 3) **page 2 statement is not clear.**
>
> Thank you very much for your comment. We have clarified this point in our revised manuscript.
>
>
> 4) **On page 5. ‘If the protected model shares a similar parameter distribution with the pre-trained model, the injected watermark could be easily erased by fine-tuning using clean i.i.d. data or adding random noise to parameters’. What is the pre-trained model? Does it mean that fine-tuning with the i.i.d data can introduce more perturbations on the watermarks than fine-tuning with OoD data? What’s the intuition behind such an claim?**
>
> Clarification for pre-trained model: The pre-trained model is a high-performance large model that requires curating a massive amount of training data from different sources, powerful computational resources, and often great efforts from human experts. Our goal is to inject watermarks to these pre-trained models for IP protection.
>
> Clarification for fine-tuning: First we want to clarify that there are two kinds of fine-tuning in our paper. One kind of fine-tuning is conducted by the model protector, which uses our generated OoD verification samples to inject watermarks. Another kind of fine-tuning is conducted by the IP infringer, which is one kind of watermark removal attack (the detailed introduction can be referred to section 2.2). For fine-tuning attack, a clean set of i.i.d or OoD data will be used to wash out the backdoor-based watermark injected by the poisoned dataset. “Fine-tuning” in your list claim is the second one. We would like to clarify by rephrasing the claim in your question: if the parameter perturbation of the protected model is small compared to the pre-trained one, the watermark will not be resilient to random noise or clean data fine-tuning. This claim is obtained from [1-2].
>
> We do not have the conclusion that fine-tuning with the i.i.d data can introduce more perturbations on the watermarks than fine-tuning with OoD data. Fine-tuning using the OoD data will usually sacrifice the model utility (standard accuracy) no matter for model protector or IP infringer. We propose a novel method in our paper to deal with this problem, and we assume an unfair scenario for the protector where the removal attack uses a part of in-distribution data to show the robustness of our proposed watermarks through extensive empirical results.
>
> We hope the above clarifications can solve your concerns about this claim.
>
> [1] Hong S, Carlini N, Kurakin A. Handcrafted backdoors in deep neural networks[J]. Advances in Neural Information Processing Systems, 2022, 35: 8068-8080.
>
> [2] Garg S, Kumar A, Goel V, et al. Can adversarial weight perturbations inject neural backdoors[C]//Proceedings of the 29th ACM International Conference on Information & Knowledge Management. 2020: 2029-2032.

---

> ### Author Response · Authors · 2023-11-17
> **Thank you for your helpful comment and suggestions - Part 2**
>
> 5) **On page 5. The statement of ‘the anomalies of the parameter distribution could be easily detected by an IP infringer’ may be conflict with the experimental results of Ref. [3]. The authors of Ref. [3] conduct experiments to check whether the distributions of model weights or model representations could be used as indicators to detect backdoors or anomalies in a DNN model.**
>
> Thank you very much for your comment. The statement in page 5 does not conflict with [3], since we have a precondition for this statement as stated in the original submission. **If the perturbation is too large**, the anomalies of the parameter distribution could be easily detected by an IP infringer [4]. Thus, we have to control the scale of the perturbation to avoid being detected. Both [3] and our proposed method show no obvious statistical difference from the original model. Thus, the backdoor is hard to be detected.
>
> [3] Programmable neural network trojan for pre-trained feature extractor.
>
> [4] Adnan Siraj Rakin, Zhezhi He, and Deliang Fan. Tbt: Targeted neural network attack with bit trojan. In Proceedings of the IEEE/CVF Conference on Computer Vision and Pattern Recognition, pp.13198–13207, 2020.
>
> 6)  **Experimental settings are not clear enough: 1) How many patches are used for training? 2) Comparison experiments are not well organized. There are many previous methods with similar strategies, e.g., BadNets, 10-invisible, smooth, Trojan Square, and Trojan watermark, and it's essential to compare with them.**
>
> 1)We investigate the effects of different numbers of verification samples in Figure 9 in the supplementary. We generate 45000 surrogate samples for watermark injection, but we do not need to use all the surrogate data for verification. We report the OoDWSR w.r.t. different numbers of samples in the verification dataset in Fig. 9. According to the figure, only 450 verification samples are enough for accurate verification.
>
> 2)6 different backdoor patterns are adopted in our experiments, including BadNets with grid, l0-invisible, smooth, Trojan Square, Trojan Square, and Trojan watermark. The watermark injection results can be found in Figure 2 in section 4.1 and Figure 5 in the supplementary.
>
> 7) **In Table 2, we observe that the OoDWSR may increase after fine-tuning and pruning. Could you give some explanations? It seems the IDWSR may also increase after such two attacks. Why? It seems to be contradicted by your previous claim.**
>
> A possible reason for the phenomenon you have mentioned is that for some of the cases, with limited clean data used for fine-tuning or pruning, the shortcut for the backdoor is well preserved, while the path in the DNN to preserve the model utility is easily corrupted (a significant ACC degradation). These results will not contradict our precious claim, since the results in Table 2 are all with weight perturbation, in this way, the backdoor is more resilient to fine-tuning or pruning attacks. The results in Table 6 show that the backdoor-based watermark without weight perturbation can be quite vulnerable to attacks, which is consistent with our previous claim.
>
> 8) **How did you implement the model extraction attack? According to the experimental results in Table 4, we still have a high OoDWSR after model extraction. Could you explain why the model extraction methods using i.i.d data can also extract the watermarks that are generated using the OoD data?**
>
> Thank you very much for your comment. We give our explanation as follows:
>
> 1)**The model extraction results shown in Table 4 are based on fine-tuning.**
>
> **Implement**: We use knockoff as an example of the model extraction attack, which queries the model to get the predictions of an auxiliary dataset, and then clones the behavior of a victim model by fine-tuning the model with queried image-prediction pairs.
>
> **Explanation**: The model extraction method will query the watermarked model for the label of the training samples, although they may get the correct label, this method will just function like clean data fine-tuning, the watermark is robust to fine-tuning as shown in our previous experiments.
>
> 2)**We also add one re-training-based model extraction in Table 8 in the supplementary.** A detailed analysis can also be found there.
>
> **Implementation**: The extracted models are re-trained from scratch. For this method, a large standard accuracy degradation (more than 10%) is observed, but the watermark is still extracted by the suspect model with high OoDWSR and p-value close to 0.
>
> **Explanation**: The backdoor information hidden in the soft label queried by the IP infringers can also embed the watermark in the extracted model. The extracted model will behave more similarly to the victim model as its decision boundary gradually approaches that of the victim model. In this way, the watermark is still extracted by the suspect model.

---

> > ### Comment · Reviewer_MV1f · 2023-11-21
> >
> > Thanks to the authors for the reply. Some of my concerns are addressed.
> >
> > However, as shown in the rebuttal, the author claims that "**our paper explores a more realistic scenario where access to ID data can be prohibitive due to data safety and confidentiality.**", I don't agree with the authors on this point. Generally speaking, most watermarks should be injected by model developers anyway, and hence they have access to all the data. The authors argue that they focus on protecting pre-trained models without training data. Even if I can accept this relatively strange setting (after all, you are protecting your own model), I cannot agree that i.i.d. data is prohibitive. The model needs to be tested after injecting watermarks and it is going to be released and used by someone, so the developers of the watermarks should have access to the data during inference, which is supposed to be i.i.d. to the training data (if not, the model doesn't deserve protection). Therefore, I don't see the problem setting justified. Moreover, the proposed solution strongly depends on (Asano & Saeed, 2023), requiring lots of augmented data for training. The required amount of data (45k for CIFAR-10) is on par with the training data size. With similar effort, we can collect the i.i.d. data and apply existing backdoor-based watermarking techniques. Consequently, it is essential to compare to the latest watermarking solutions.

---

> ### Author Response · Authors · 2023-11-21
>
> Thank you very much for your response. We agree that the traditional setting with ID data is applicable to some of the scenarios, but for the ID data-free problem setting, we list two realistic scenarios in the paper. (1) For example, someone trying to protect a model fine-tuned upon a foundation model and a model publisher vending models uploaded by their users. Another example is an independent IP protection department or a third party that is in charge of model protection for redistribution. (2) Yet another scenario is federated learning where the server does not have access to any in-distribution (ID) data, but is motivated to inject a watermark to protect the ownership of the global model. The traditional methods cannot handle these two scenarios. Watermark injection without training data is barely explored. This paper focuses on tackling this challenge.
>
> For the efforts of data collecting, we only have to collect one OoD image, and then do the data augmentation (the augmentation can be done within a few seconds), instead of collecting 45k i.i.d images, which is much more sample efficient.
>
> In our paper, we also compare with ID data watermark to verify the robustness of our method in Table 3. According to the results, after RT-AL, the watermark success rate drops to 4.13% and 3.42%, respectively for ID poison (baseline), while drops to 57.52% and 24.19% for OoD poison (proposed method), which verifies that our proposed method is also much more robust against watermark removal attacks.

---

> > ### Comment · Reviewer_MV1f · 2023-11-22
> >
> > I agree that the training data-free problem setting is valid in certain scenarios, but this is not equivalent to ID data-free. You have to have ID data to test and use the model, so there is no such requirement for ID data-free watermarking.

---

> > > ### Author Response · Authors · 2023-11-22
> > >
> > > We appreciate your agreement on training data-free, but we want to argue the ID data-free scenario does exist for federated learning (FL) setting [1], where multiple local clients cooperatively train a high-quality global model without sharing their ID data. In FL, the server does not have access to any ID data,  since ID data belongs to the individual clients due to data safety and confidentiality requirements. Thus, the server has to inject a watermark to protect the ownership of the global model in an ID data-free way.
> > >
> > > Another reason we want to propose this ID data-free method is that collecting so much ID data is time-consuming and laborious. Collecting just one single OoD image is much more sample-efficient. For the usage of the model as you have mentioned, the model users can only test very few samples they have collected, but this limited number of ID samples is not enough for watermark injection. In addition to this, not all model users will use the pre-trianed model for testing ID samples. Some of the model users will even use the pre-trained model to test downstream tasks for OoD samples.
> > >
> > > Besides, we also verify in Table 3 in the experiments that our method is more robust compared with ID watermark injection.
> > >
> > > [1] Jakub Konecny, H Brendan McMahan, Felix X Yu, Peter Richtárik, Ananda Theertha Suresh, and
> > > Dave Bacon. Federated learning: Strategies for improving communication efficiency. arXiv
> > > preprint arXiv:1610.05492, 2016

---

### Official Review · Reviewer_6zjD · 2023-11-01

**Soundness:** 2 fair
**Presentation:** 3 good
**Contribution:** 3 good
**Rating:** 6
**Confidence:** 2

**Summary:**

The paper proposes a novel method for watermarking deep neural networks to protect their intellectual property and commercial ownership. Traditional watermarking strategies involve adding backdoor triggers to training samples, but these methods have limitations in terms of data privacy, safety concerns, and vulnerability to model changes. In contrast, the proposed technique leverages the knowledge from a single out-of-distribution image as a secret key for IP verification. The proposed method can immune to third-party promises of IP security. The watermark is injected by perturbing the model parameters randomly, which enhances robustness against common watermark removal attacks such as fine-tuning, pruning, and model extraction. The experimental results show that the proposed watermarking approach works well without requiring additional training data. Furthermore, it demonstrates robustness against the watermark removal attacks.

**Strengths:**

IP Protection: The paper addresses the challenge of protecting the intellectual property of deep neural networks, which is increasingly crucial in the field. It has attracted increasing attentions in the literature, especially in the era of foundation models. This paper considers an important problem, potentially having large impact to the community.

Safe and Robust Technique: The proposed technique claims to be safe and robust. It leverages a single out-of-distribution (OoD) image as a secret key for IP verification. This approach avoids privacy and safety concerns associated with poisoning training samples. By inducing random perturbations of model parameters during watermark injection, it aims to defend against common watermark removal attacks such as fine-tuning, pruning, and model extraction.

Agnostic to Third-party Promises: The technique is described as agnostic to third-party promises of IP security. This suggests that the proposed approach does not rely on external entities or services for ensuring IP protection, which can be beneficial in scenarios where trust in third parties is limited.

Time- and Sample-Efficient: The proposed watermarking approach is claimed to be time- and sample-efficient without requiring additional training data. This could be advantageous as it reduces the computational resources needed for watermarking.

**Weaknesses:**

Training time watermarked data will have some distribution discrepancy over test situations, thus one cannot ensure the performance of watermarking. It seems that such an issue remains an open question in IP protection, so more discussion about "how to handle such a problem" or "is it an important issue in the literature" can be formally discussed.

Do more data points, other than just one point, will lead to more contributions to the watermarking strategy? Experimental verification and heuristic explanation seem to be interesting. More ablation studies, such as number of data points, hyper parameter setting, choice of validation set, and effects of individual modules, can be presented, which can largely improve the solidity of this paper.

For the first equation in Sec 3.1, I am not sure if the first term is contributive. In previous works of watermarking, the first term is used to ensure the original high performance for the original task (I guess). Therefore, from my view, the first term is not important, especially considering the fact that x does not follow the same distribution of original data distribution, and no semantic information should be kept therein.

A related question  is that does the watermarking strategy suffer from catastrophic forgetting? In other words, does learning from tilde D impact the performance for the original task, especially for foundation models. More discussion and empirical justification should be given here. Another related issue is that, it seems that this paper is motivated by the recent progress in foundation models, but it seems that this paper does not provide enough evaluation for the power of watermarking with backbones of foundation models.

In previous works, researchers demonstrate that weight perturbation can lead to data transformation [1]. Therefore, the reason  why the suggested method works may also be explained from the perspective of implicitly introducing more data.  Another related question is that if introducing more data can also mitigate the impact of watermarking removing. More discussion and evaluation should be considered here.


[1] Relative Flatness and Generalization

**Questions:**

Please see the weaknesses above.

---

> ### Author Response · Authors · 2023-11-17
> **Thanks for your helpful comments and suggestions - part 1**
>
> We are glad that the reviewer found our problem setting important and our method advantageous. We thank the reviewer for the constructive comments and suggestions, which we address below:
> 1) **Training time watermarked data will have some distribution discrepancy over test situations, thus one cannot ensure the performance of watermarking. It seems that such an issue remains an open question in IP protection, so more discussion can be formally discussed.**
>
> We would like to thank the reviewer for the question.
> For watermark verification, there is no distribution discrepancy between the data used for watermark injection and watermark verification, since they adopt the same trigger set (verification set).
>
> For the main standard accuracy test on the main task, although the poisoned dataset we generated for watermarking has a different distribution with ID samples, a recent study [1] has shown that using the augmented OoD images for training a classifier yields reasonable performance on the main prediction task. Thus, we conjecture that it is plausible to inject backdoor-based watermarks efficiently to different parts of the pre-trained representation space by exploiting the diverse knowledge from one single OoD image. In this way, we can not only maintain the model utility but also successfully inject the watermark.  We discussed this problem at the beginning of section 3. Our extensive experimental results also verify our intuition.
>
> 2) **Do more data points, other than just one point, will lead to more contributions to the watermarking strategy? More ablation studies can be presented, which can largely improve the solidity of this paper.**
>
> Thank you very much for your suggestion for these ablation studies. We add one experiment that uses more than one OoD image to generate poisoned datasets in Table 10 in the supplementary. Although the watermark with more OoD images is not more robust against fine-tuning conducted by the IP infringer, using multiple OoD images will make it harder for the IP infringers to get access to the source OoD image and guess the composition of our validation set, which can further enhance the security of the watermark.
>  We also show the effect of choosing different OoD images in Table 5 in section 4.4.  We also investigate the effects of different numbers of verification samples in Figure 9 in the supplementary. The effect of backdoor weight perturbation is shown in Table 6 in section 4.4, and Table 7, Figure 6 in the supplementary.
>
> 3) **For the first equation in Sec 3.1, I am not sure if the first term is contributive. In previous works of watermarking, the first term is used to ensure the original high performance for the original task. Therefore, from my view, the first term is not important.**
>
> The first term is also used to ensure high performance for the original task as in previous works. The intuition is from recent studies [1-2], which showed a surprising result that one single OoD image is enough for learning low-level visual representations provided with strong data augmentations. Previous work has shown that using OoD images for training a classifier yields reasonable performance on the main prediction task [1]. We explain the intuition at the beginning of section 3, and also add the explanation for the equation in the revised manuscript to avoid misunderstanding.
>
> 4) **A related question is does the watermarking strategy suffer from catastrophic forgetting? In other words, does learning from tilde D impact the performance for the original task, especially for foundation models. More discussion and empirical justification should be given here. Another related issue is that, it seems that this paper is motivated by the recent progress in foundation models, but it seems that this paper does not provide enough evaluation for the power of watermarking with backbones of foundation models.**
>
> The proposed watermarking will not impact the performance of the original task, since we can control the standard accuracy degradation less than 3\% as shown in the experiment part. The first term for the first equation of sec 3.1 is used to ensure high performance for the original task as mentioned in the previous question.   We only discuss watermarking for vision models in this paper. We agree that ID data-free watermarking for foundation models can be an interesting and open problem, which we leave for our future work.
>
>
> [1] Extrapolating from a single image to a thousand classes using distillation. In ICLR, 2023.
>
> [2]  A critical analysis of self-supervision, or what we can learn from a single image. arXiv, 2019

---

> ### Author Response · Authors · 2023-11-17
> **Thank you for your helpful comment and suggestions - Part 2**
>
> 5) **In previous works, researchers demonstrate that weight perturbation can lead to data transformation [3]. Therefore, the reason why the suggested method works may also be explained from the perspective of implicitly introducing more data. Another related question is that if introducing more data can also mitigate the impact of watermarking removing.**
>
> We appreciate your insight on the connection of our method and introducing more data. [3] investigated an interesting problem that the input data perturbation can also be transformed into the parameter space perturbation, which also inspired us a lot. Both input (feature) perturbation and parameter space perturbation can lead to the loss flatness, which improves the model’s generalization ability.
>
> Our method of adding weight perturbation to improve the robustness of the watermark is more intuitive. Fine-tuning based watermark removal attacks will modify the model parameter space with small perturbations, thus, we also try to improve the robustness of the watermark by adding perturbations to the parameter space.
>
> We agree with you that adding data perturbation is another interesting perspective. But directly increasing the dataset size may not work, since how similar these trigger set data are, will they be so homogenized that they will serve little purpose for perturbation remains a problem. What is the best way to generate perturbations for the trigger set is a heuristic problem. [3] is typically designed for generalization instead of backdoor/watermark, thus, how to improve the backdoor robustness using data perturbation is beyond the scope of this paper. We are inspired to explore all these problems in our future works.
>
> [3] Relative Flatness and Generalization

---

> ### Author Response · Authors · 2023-11-21
> **A kind reminder to reviewer 6zjD**
>
> Dear Reviewer 6zjD,
>
> Thank you for your time to review our paper and leave valuable comments and suggestions. We are wondering whether you have got a chance to read our response to your questions. We will be glad to provide more explanations and answer more questions if you have any.
>
> Authors

---

### Meta-Review · Area_Chair_y9o7 · 2023-12-08

**Metareview:**

This paper proposes a watermark injection technique independent of training data, thereby achieving enhanced data privacy and safety.

During the rebuttal, reviewers highlighted several strengths. Specifically, 1) a safe and robust technique; 2) time and sample efficiency; 3) a novel method.

During the rebuttal, one major concern was raised by Reviewer MV1f regarding the motivation behind this approach, specifically for scenarios involving a lack of training data during watermark injection. This scenario may be limited.

However, there are arguments during the internal discussions. Reviewer URnu commented that the concern should be considered minor. Specifically, some real-world scenarios, e.g., federated learning, have been illustrated. Reviewer MV1f also acknowledges that such real-world scenarios. The AC agrees that the proposed method works for the mentioned real-world scenarios.

The AC therefore recommends accepting this paper. However, the authors are strongly encouraged to revise the paper to add more real-world scenarios and discussions according to the advice from Reviewer MV1f. It can help emphasize the practical significance of the paper.

**Justification For Why Not Higher Score:**

NA

**Justification For Why Not Lower Score:**

Several key strengths have been highlighted by reviewers, including 1) a safe and robust technique; 2) time and sample efficiency; 3) a novel method. The major concern about the motivation for the lack of training data during watermark injection, raised by Reviewer MV1f, has been sufficiently addressed.

---

### Decision · Program_Chairs · 2024-01-16

Accept (poster)